# Comparative Study of Heat- and Enzyme-Induced Emulsion Gels Formed by Gelatin and Whey Protein Isolate: Physical Properties and Formation Mechanism

**DOI:** 10.3390/gels8040212

**Published:** 2022-03-31

**Authors:** Siqi Li, Guipan Chen, Xinyue Shi, Cuicui Ma, Fuguo Liu

**Affiliations:** College of Food Science and Engineering, Northwest A & F University, Yangling, Xianyang 712100, China; siqisweet@nwafu.edu.cn (S.L.); chenguipan0717@163.com (G.C.); 2021Sxy@nwafu.edu.cn (X.S.); guocui2007@163.com (C.M.)

**Keywords:** emulsion gels, texture, rheology, tunable, design principles

## Abstract

Emulsion gels have received increasing attention due to their unique physicochemical properties. In this paper, gelatin and whey protein isolate (WPI) were used to construct emulsion-filled gels by heat-induced or enzyme-induced methods, and their rheology, texture properties and microstructure were explored and compared. The effect of the preparation methods, emulsion droplet characteristics and gel matrix concentration on the elastic modulus and hardness of the gels were firstly investigated, then the key control factors were picked out by calculating the Pearson correlation index, and the design principle was constructed by combining these factors flexibly for emulsion gels with adjustable texture. The results show that the emulsion gels formed by different preparation methods have completely distinct microstructures and emulsion distributions, as well as the macroscopic properties of the gels, specifically the enzyme-induced gels exhibited greater elastic modulus and hardness, while heat-induced gels were softer and more delicate. In addition, the droplet sizes of filled emulsions and matrix concentration mainly affected the rheological properties and hardness of the gels. This study successfully established the design principles of emulsion gels with tunable texture structure, which provided a reference for targeted gels preparation according to the texture properties required by specific application scenarios.

## 1. Introduction

Emulsion gels are a complex colloidal material composed of emulsion and gels. Due to its special emulsion distribution and three-dimensional network structure, emulsion gels have a wide range of applications in the food industry, such as delivery vehicles for hydrophobic bioactive ingredients, development of low-fat products or as fat substitutes [1,2,3]. According to the kind of food materials constituting the gels matrix, emulsion gels can be divided into protein, polysaccharide and protein-polysaccharide emulsion gels. Especially, protein emulsion gels are commonly found in people’s daily diet, such as coagulated yogurt and cheese based on milk protein, tofu and processed meat products made from soybean protein, and people often eat these gelatinous foods to meet needs of nutrition and health in daily life [4]. In addition, the formation of gels is an ideal approach to improve and impart the texture characteristic of foods [5]. Generally, the quality of gels is reflected in textural properties such as hardness, viscoelasticity, water holding capacity and microstructure [4]. There are many factors that affect the texture properties of emulsion gels, including the type of raw materials, gelation methods and the addition of other excipients [6,7].

Depending on the application conditions, gel-forming methods can be classified into physical, chemical and enzymatic methods. The physical method is carried out by heating, adding acid, salt or other excipients; the chemical method is to promote chemical reactions between proteins through chemical cross-linking agents such as genipin or glutaraldehyde; and the enzymatic method can prompt the covalent cross-links between proteins under the catalysis of biological enzymes such as laccase, transglutaminase (TGase/TG) and tyrosinase [4,8,9,10,11]. Compared with the chemical method which may have safety problems, the physical and enzymatic methods have received more extensive attention in the preparation of food-grade emulsion gels [3,12,13]. Heat treatment is the most convenient and direct physical method. For globular proteins, the formation process of heat-induced gels can be summarized as the protein structure unfolding because of heat denaturation, while accompanied by the exposure of hydrophobic groups, and then the molecules interact with each other closely to form aggregates and finally to form gels [6,10,14]. Moreover, enzymatic method is a safe and green gelation means with mild action conditions, which can effectively catalyze chemical reactions between polymers [13,15,16].

Generally, two or more biopolymers as raw materials to prepare gels can often endow the system with better mechanical properties and texture structure, making up for the lack of a single polymer gel network [17]. An emulsion gel composed of whey protein isolate (WPI) and sodium alginate with self-supporting properties using double cross-linking of TGase and calcium ions was successfully constructed [7]. The dual polyphenol-loaded composite emulsion gels constructed by zein and sodium alginate significantly improved the photostability and bioavailability of curcumin and resveratrol [18]. The emulsion-filled gels constructed with wheat bran arabinoxylan and soybean protein isolate had excellent water holding capacity and achieved the effective sustained release of β-carotene compared with a single gel [3].

In this study, WPI and gelatin were used as the matrix materials to prepare emulsion gels. WPI is widely utilized as ingredients in food processing due to their high nutritional value and good physical properties such as emulsifying and gelling properties, which could be used for the improvement of food texture and the development of new products [19,20]. Gelatin has abundant surface-active groups and excellent functional properties, and it is used as a thickener, gelling agent and emulsifier in food industry [21,22,23]. These two proteins were firstly mixed, and an oil-in-water emulsion was dispersed into a WPI-gelatin solution, then the mixture was induced to gel by a heating and enzymatic method. Following this, the differences in physical properties of gels under different cross-linking methods were comprehensively compared and the possible factors that may affect the texture structure of emulsion gels were explored, then the correlation law between the gel preparation conditions and the texture characteristics was also investigated; on this basis, the design principle of tunable emulsion gels texture structure was established. This paper will provide a scientific basis for improving food texture and designing gelatinous foods with suitable texture characteristics.

## 2. Results and Discussion

### 2.1. Formation of WPI-Stabilized Emulsions with Different Interfacial Properties

The optimum concentration of WPI was screened by droplet size and zeta potential (as shown in Figure 1). Emulsion droplet size decreased significantly (*p* < 0.05) with the increase in WPI concentration, which may be related with the adsorption of more WPI molecules in the continuous phase to the oil–water interface, thus reducing bridging flocculation between droplets. When WPI concentration increased to 2.0 wt%, droplet size did not change significantly (*p* > 0.05). Similarly, zeta potential (absolute value) increased significantly with the increase in WPI concentration (*p* < 0.05), which was consistent with the previous report, because more negative net charges of WPI molecules adsorbed on the surface of oil droplets [24].

In order to obtain emulsions with different droplet sizes, the homogeneous pressure was varied to 5, 10 and 50 MPa at fixed cycle numbers under a WPI concentration of 1.5 wt% [25]. It can be seen from Figure 1c that emulsion droplet size was significantly different (*p* < 0.05) after homogenization at different pressures, which were 1200 nm, 650 nm and 350 nm, respectively.

### 2.2. Formation of Emulsion Gels with Different Rheology and Texture Properties

#### 2.2.1. Effect of Emulsion Filling Volume

The influence of emulsion filling volume on the storage modulus and hardness of gels is shown in Figure 2a–c. When filling volume was 5% and 10% *v*/*v*, the storage modulus of TGase-induced gels was higher than the emulsion blank group and had the highest storage modulus at the volume of 10% *v*/*v*, indicating that emulsion droplets as an active filler could enhance the elasticity of gels [26]. When the filling volume was 15% *v*/*v*, the storage modulus was lower than the emulsion blank group, which may be because the existence of too many emulsion droplets partially hindered the cross-linking of TGase to the gel matrix. It is shown in Figure 2c that the hardness of emulsion gels at 10% *v/v* filling volume was higher than the emulsion blank gel, and 15% *v/v* filling volume has the lowest hardness, which was consistent with the results of rheology, indicating that the texture of the gels was weakened when emulsion acted as an inactive filler.

For heat-induced gels (Figure 2b), 5% *v/v* emulsion filling increased the storage modulus compared to the emulsion blank gel, while storage modulus decreased to various extents at the filling of 10% *v/v* and 15% *v*/*v*. In addition, the hardness of gels decreased significantly (*p* < 0.05) as the filling volume increased from 0% to 10% *v/v* (Figure 2c). When heated at 85 °C for 30 min, WPI unfolded and hydrophobic groups were exposed, meanwhile gelatin was a liquid condition [27]. Then, rapidly cooled to room temperature and cured at 4 °C, WPI molecules formed larger aggregates, and gelatin changed from a disordered state to an ordered spiral structure, and this cooling aggregation process was relatively slow compared with TGase induction [28]. With the increase in the emulsion filling volume, the number of oil droplets dispersed in the matrix increased, so oil droplets impeded the interaction and aggregation among protein molecules due to the steric hindrance effect, and the effective matrix concentration in the aqueous phase was decreased which also hindered the formation of a gel network, causing the gels hardness to decrease gradually.

It should be noted that for two gelation methods, filling the same volume of emulsion could facilitate a reasonable comparison in the follow-up experiments. Considering the above results, 10% *v/v* emulsion filling volume was selected for the subsequent experiment.

#### 2.2.2. Effect of Emulsion Droplet Size

Besides the filling volume, there were other emulsion characteristics such as droplet size which also may affect the rheology and texture of emulsion-filled gels [25]. Figure 2d–f shows the effect of emulsion droplet size on the storage modulus and hardness of the gels. It is shown clearly in Figure 2d that the storage modulus of TGase-induced gels was affected by emulsion droplet size, and the gel filled with 650 nm emulsion had the highest storage modulus, followed by the 350 nm and 1200 nm emulsions. Likewise, the addition of these three sizes of emulsions to the hydrogel matrix resulted in stiffer gels compared to the emulsion blank group, indicating that these emulsions acted as active fillers to enhance the storage modulus and hardness of the gels at 10% *v/v* filling volume, and when droplet size was too large, the enhancement effect was less; in contrast, the emulsion with a smaller size could bring a more obvious enhancement which is attributed to a higher surface area to volume ratio [29]. Moreover, when the volume fraction of the oil phase in the emulsion was the same, during the process of homogenization, the higher the pressure, the more obvious the breaking effect of oil droplets, which accompanied the smaller droplet size and more droplets, so the distance between droplets was lessened and emulsions dispersed more closely in the gel matrix for greater enhancement impact [30,31].

For heat-induced samples, the gel had the largest storage modulus when the droplet size was 1200 nm, which was higher than the emulsion blank gel, while the storage modulus of 650 nm and 350 nm emulsion gels decreased in turn and was smaller than the emulsion blank gel. It is shown that the role of filler could be adjusted under the help of changing the droplet size when the emulsion filling volume was fixed. Similarly, when the filling volume was constant, the smaller the droplet size and the more droplets in the emulsion, and the number of oil droplets dispersed in the unit volume network increased correspondingly. After heating treatment, more oil droplets would accumulate to form aggregates, resulting in the surface area between droplets and protein molecules also increasing, which may lead to a greater steric hindrance, prevent the exposure and aggregation of hydrophobic groups of proteins, and finally, weaken the elastic properties and hardness of the gels network.

#### 2.2.3. Effect of Gelatin Concentration

It was reported that modulus of the gel matrix affected the rheological properties of emulsion gels, and the storage modulus of the gel matrix was closely related to the polymer concentration in the matrix [29]. The impact of the matrix concentration on emulsion gels rheology and texture was investigated, and the results are shown in Figure 2g–i. The storage modulus and hardness of all gels increased with the increase in gelatin concentration, indicating that the polymer concentration in the continuous phase played an important role in improving the rheology and texture of emulsion gels. When more protein molecules were involved in construction, the gels network became more compact. This result was consistent with the previous research that increasing the gelling agent concentration brought polymer chains closer together, thus reducing the number of voids in the matrix and creating more intermolecular interactions [29,32,33]. Moreover, when the gelatin concentration was identical, the storage modulus and hardness of TGase-induced gels were much higher than those of heat-induced gels. In addition, compared with TGase induction, the heat-induced gels had more obvious frequency dependence, suggesting the force of maintaining network formation was different using the two methods. TGase could induce irreversible covalent cross-linking, while the heat-induced gels were mainly formed by hydrogen bond, hydrophobic interaction and disulfide bond [34].

### 2.3. Design Principles for Texture-Tunable Emulsion Gels

The aforementioned results showed that the rheology and texture of the emulsion gels were affected by the emulsion filling volume, droplet size and gelatin concentration. Therefore, the most important factors affecting gels properties were picked out based on the Pearson correlation coefficient among these variables, and the results are shown in Table 1. Under both gelation methods, there was a significant positive correlation between the gelatin concentration and the storage modulus or the hardness of the gels (r > 0.990, *p* < 0.01). The filling volume and droplet size of TGase-induced gels were negatively correlated with the storage modulus and hardness. For heat-induced gels, there was a significant positive correlation (r = 1.000, *p* < 0.05) between droplet size and hardness. Thus, the gelatin concentration and emulsion droplet size were the foremost factors affecting the rheology and texture of gels.

To obtain the emulsion gels with a different texture and structure, two gelation methods, gelatin concentration and emulsion droplet size were selected as variables. It should be noted that the comparative study of the two gelation methods ran through the whole experiment. Here, the design principle of emulsion gels was a flexible combination based on the two foremost factors, and the combination method can be summarized as ‘strong with strong, weak with weak’, that is, the droplet size with the smallest storage modulus filled with the lowest gelation concentration. To distinguish these samples, TGase-induced gels were recorded as A and heat-induced gels as B, and numbers following the letters from small to large corresponded to the gelatin concentration from small to large. The specific gel samples are listed below: A1(2 wt%-1200 nm), A2 (3 wt%-350 nm), A3 (4 wt%-650 nm), B1 (2 wt%-350 nm), B2 (3 wt%-650 nm) and B3 (4 wt%-1200 nm).

### 2.4. Analysis of Gelation Mechanism of Emulsion Gels

The molecular weight (MW) changes of the two proteins in the gel matrix were observed by SDS-PAGE, and the information about the forces supporting gels formation was obtained [35]. In this part, A3 and B3 gels were selected as the representatives formed by the two gelation methods, and they were both treated by reduction and non-reduction, respectively.

Figure 3a shows the results obtained after the reduction treatment. Lane 1 was native WPI, where two bands at the bottom corresponded to the main components of it, namely β-lactoglobulin (β-Lg, 18.5 KDa) and α-lactalbumin (α-La, 14.2 KDa), and the former was wider than the latter, indicating that β-Lg occupies a large proportion in WPI, which was consistent with a previous study [36]. Lane 2 was original gelatin and multiple bands appeared, indicating it was composed of non-uniform polypeptide chains with a molecular weight distribution range of 35–180 KDa. It was reported that Type A gelatin had a wider molecular weight distribution than Type B, and the molecular weight distribution range was 53–220 KDa, which was larger than the results obtained in this experiment, and this difference may be related to the concentration of the separating gels used in PAGE and the different staining methods [37]. Not only were β-Lg and α-La retained in B3 (Lane 3), but also the gelatin bands higher than 35KDa were retained, while there were no obvious bands above 35 KDa except for the lighter β-Lg and α-La bands for A3 (Lane 5) gel. It was suggested that the WPI gel contained in B3 was completely destroyed by the reducing loading buffer, and the gelatin gel was also disassembled into polypeptide chains under the action of the buffer which was similar with the native condition. In contrast, only the part of the structure in WPI gel contained in A3 was damaged by the loading buffer, and the rest of the WPI and all gelatin were covalently cross-linked under the catalysis of TGase, and thus still existed in a cross-linked state instead of polypeptides.

The result of the non-reduced PAGE is shown in Figure 3b. The bands at 18.5 KDa and 14.2 KDa in B3 nearly disappeared, but there were still multiple bands of gelatin above 35 KDa, while almost no bands appeared in A3. It was indicated that the formation of WPI gel in B3 mainly depended on the formation of disulfide bonds; similarly, the WPI gel in A3 also partially involved disulfide bonds which were attributed to the pre-heating treatment. Furthermore, the presence or absence of gelatin bands in B3 and A3 illustrated clearly that TGase exhibited efficient cross-linking on gelatin, while there was no similar covalent cross-linking between the heat-induced gelatin molecules.

Based on the above results, it was found that under the heat treatment, the most important force maintaining WPI gel was the intermolecular or intramolecular disulfide bonds. For the TGase induction, gelatin molecules formed network links through covalent bonds by an acyl transfer reaction, while there were not only enzymatic covalent crosslinks between molecules, but also a part of the network was formed by disulfide bonds in the formation of WPI gel.

### 2.5. Microstructure of Emulsion Gels

#### 2.5.1. CLSM Observation

The physicochemical properties of emulsion gels are inseparable from their structure. To obtain more information on the structural differences of the gels formed under the two gelation methods, the structural distribution was first observed by CLSM.

It is shown in Figure 4A1–A3 that the TGase-induced emulsion gels were smooth, flat and off-white in appearance with self-supporting properties, and there was no obvious difference in appearance among the three samples. CLSM images show that green oil droplets were independent and uniformly dispersed in the gel matrix, while the red gel matrix was continuous and dense, additionally, the increase in gelatin concentration did not cause changes in the matrix structure. Larger oil droplets were observed in Figure 4(A1a), while the droplets were smaller and more numerous in Figure 4(A2a), which was accordant with the droplet size of the filled emulsion. In addition, there were scattered circular black voids in the red matrix, and these voids were distributed in the same position as the green oil droplets, suggesting that the emulsion was successfully filled into the continuous phase and occupied a certain space.

The heat-induced gels (Figure 4B1–B3) also had a self-supporting property, and their surfaces were smooth and delicate with a bright white color. The size of the oil droplets was basically consistent with the initial size of the filled emulsions, but a small amount of the droplets aggregated after heat treatment, which may be related with the process of gels formation. However, since the essence of CLSM was still an ordinary optical microscope, the magnification was limited, meanwhile the droplet size of the homogenized emulsions was nanoscale, thus no obvious oil droplet flocculation was observed in this field of view. Moreover, the distribution of the red gel matrix was not continuous and showed granular aggregated accumulation.

#### 2.5.2. Cryo-SEM Observation

The three-dimensional network structure inside the gels was not directly observed after only staining with fluorescent dyes. In recent years, Cryo-SEM has been successfully used to observe products with high water and fat content, such as emulsion gels, curds and cheese [38]. Therefore, Cryo-SEM was selected in this study to observe the three-dimensional structure of emulsion gels.

It can be seen from Figure 5a that A3 gel had a uniform and porous network structure, and the part marked by the yellow circle was the emulsion oil droplets dispersed in the network, while several droplets were dispersed independently in the field of view. An oil droplet on the left was in a complete spherical shape, while the droplet on the right was only half and showing a hemispherical cross-section, because the gel sample was interrupted before observation. Taking the oil droplet in the yellow circle on the right as the center of the field of view, a clearer image was obtained by local magnification, namely Figure 5b. The network structure formed by the catalysis of TGase induction was thicker and had a compact structure; simultaneously, a connecting part was observed between the surface of the oil droplet and the surrounding matrix, indicating that the WPI adsorbed on the oil–water interface, interacted with the protein molecules in the gel matrix and jointly participated in the construction of the network, so acted as an active filler and enhanced the rheological and textural properties of the gels.

The structure of the B3 gel is shown in Figure 5c, and it was found to be completely different from the A3 gel. There were a large number of oil droplets in the field of view, and they were a larger size and in an aggregated state. The gel network, existing in the place without oil and in the gaps between adjacent droplets, was not continuous due to the blocking of oil droplets. Compared with A3 gel, the pores of the B3 were smaller and denser, while the pore walls were thinner and discontinuous. At a higher magnification (shown in Figure 5d), it was observed that the adjacent oil droplets form connections through the surface-matrix-surface, indicating that the emulsifier WPI in the emulsion interacted with the matrix during the heat treatment, and participated in the construction of gel network.

### 2.6. Textural Properties of Emulsion Gels

#### 2.6.1. Hardness Analysis

The textural properties of these six emulsion gels were determined. As can be seen from Figure 6a, the hardness of A1, A2 and A3 gels had significant difference (*p* < 0.05), which was attributed to the increase in the effective matrix concentration that participated in the network construction due to the increase in the gelatin concentration. Similarly, the hardness of B1, B2 and B3 gels was also significantly different (*p* < 0.05). Compared with increasing the gelatin concentration individually (Figure 2i), simultaneously changing the emulsion droplet size and gelatin concentration could result in a higher hardness, indicating that the effect of modulating the droplet size and matrix could control the gels texture effectively. In addition, the TGase-induced gels had higher hardness than the heat-induced gels at the same gelatin concentration, showing that the network based on ε-(γ-glutamine) lysine isopeptide was stronger and firmer than that maintained by disulfide bond, hydrophobic interaction and hydrogen bond [27,39].

#### 2.6.2. Stress-Relaxation Analysis

The stress-relaxation curves of emulsion gels are displayed in Figure 6b,c. When the strain was kept constant at 10% for 60 s, the stress decreased slowly and tended to level off with the extension of time for TGase-induced gels, while the stress of heat-induced gels showed a tendency of decreasing slightly at first, then changing slowly and finally, tending to smooth, indicating that the emulsion gels had stress relaxed phenomena and was a semi-solid material having both elastic and viscous.

In order to describe the stress relaxation process of viscoelastic emulsion gels better, the raw data of relaxation phase was fitted with the Maxwell three-element rheological model, which consisted of a spring in parallel with a Maxwell Model. The mathematical expression is:(1)$$F\left(t\right){=D}_{0}{\cdot E}_{0}{+D}_{0}{\cdot E}_{1}\cdot \exp(-t/T)$$

Among them, F(t) means the load (N), t is the arbitrary relaxation test time, D_0_ is a certain amount of deformation (mm), E_0_ is the equilibrium elastic coefficient (N/mm), E_1_ is the decay elastic coefficient (N/mm), T is the relaxation time (s), η is the damping body viscosity coefficient (N·s/mm), η = T·E_1_.

The four model parameters including E_0_, E_1_, T, η reflect the relaxation characteristics and texture of the material and the results of nonlinear fitting are shown in Table 2. The determination coefficients R^2^ of all samples were between 0.96–0.98, indicating the relaxation characteristics of emulsion gels was reflected well using this model. The equilibrium elastic modulus of TGase-induced gels was higher than that of heat-induced gels, suggesting TGase-induced gels had higher elastic properties and the new balance could be achieved in a short time through the adjustment of the intermolecular distance and the entanglement coupling of the macromolecular polymer chain, all the while releasing hydraulic pressure [40]. The relaxation time and damping body viscosity of heat-induced gels were both greater than TGase-induced gels, which illustrated that their internal structure was more viscous.

### 2.7. Rheological Properties of Emulsion Gels

The rheological properties of emulsion gels were investigated including frequency and flow scanning. As shown in Figure 7a, six emulsion gels had different storage modulus and displayed a growing trend with the increase in gelatin concentration, while the storage modulus of TGase-induced gels were much larger than the heat-induced one, showing that the former was more elastic while the latter was softer, which corresponded to the direct touch feeling exhibited by its macroscopic. Compared with the result in Figure 2g, the storage modulus of A1, A2 and A3 gels decreased though they had the same gelatin concentration, indicating that, besides changing the matrix concentration, the alteration of droplet size would also affect the texture of gels. The impact of droplet size not only affects the number of oil droplets per unit volume of emulsion, but also affects the cross-linking and enzymatic sites through steric hindrance. However, the difference in gels texture caused by emulsion droplet size was less obvious than the change in polymer concentration, which provided a reference for the design of texture according to the specific applications of different food systems.

As shown in Figure 7b, the apparent viscosity of all samples decreased rapidly with the increase in shear rate until it became stable, which named shear thinning behavior, indicating the emulsion gels was pseudoplastic. With the increase in gelatin concentration, the apparent viscosity also increased. Except for gels at 4 wt% gelatin, the apparent viscosity of the TGase-induced gels was larger than the heat-induced gels with the increase in shear rate, which was attributed to the magnitude difference in storage modulus between the two. When the gelatin concentration was 4 wt%, the proportion of gelatin in the gel matrix was as high as 40%. The higher concentration of polymer increased the effective concentration of the gel network. Even though the progress of heat-induced gels was relatively slower, a higher gelatin concentration and larger droplet size promoted the steric hindrance caused by the aggregation of oil drop decrease to a great degree. More importantly, the initial pH of WPI was closer to the isoelectric point, which was in favor of the WPI aggregated and formed a network effectively, then participated in the co-construction of a composite gel. Because of the effective participation of WPI, its own characteristics brought a more delicate and viscous gel texture.

### 2.8. Hydration Properties of Emulsion Gels

#### 2.8.1. Water Holding Capacity

As shown in Figure 8a, the WHC of heat-induced gels increased significantly with the increase in gelatin concentration (*p* < 0.05), indicating that the increase in matrix polymer concentration benefitted the formation of more compact and firmer gels. In addition, when the gelatin concentration was 4 wt%, the emulsion with the largest droplet size was filled so that the steric hindrance, because of oil droplets flocculation on protein aggregation, was relatively less due to the decreased droplet numbers. Under the common influence of both, the B3 gel had the highest WHC, followed by B2 and B1. In contrast, there was no significant difference (*p* > 0.05) in the WHC of TGase-induced gels though the gelatin concentration increased. Overall, regardless of the gelation methods, emulsion gels always obtained a higher WHC (>90%), and heat-induced gels were better than the TGase-induced gels, on account of the two proteins involved in gels construction having good hydrophilicity. On the other hand, appropriate preparation parameters were obtained with our efforts in previous pre-experiments that included exploration such as enzyme dosage, pH of WPI and incubation time.

#### 2.8.2. Swelling Ratio

As shown in Figure 8b, the swelling ratio of heat-induced gels had no significant difference (*p* > 0.05) and was smaller than the TGase-induced gels which decreased significantly with the increase in gelatin concentration (*p* < 0.05). Based on the above, we found an interesting phenomenon, the increase in gelatin concentration caused the WHC of the heat-induced gels to increase and the swelling ratio of the TGase-induced gels to decrease. For the TGase-induced gels, the filled emulsion could be quickly fixed in a three-dimensional network and distributed uniformly. When the volume of the pre-gel mixture was identical, the higher gelatin concentration meant more protein molecules, so the smaller intermolecular distance and denser network generated by enzyme catalysis resulted in the outside water needing to overcome more resistance to enter the gels. At this moment, the original internal space was occupied by more connected macromolecules, so there was less new water that can be accommodated; instead, for the lower gelatin concentration, molecular spacing was relatively far and the gel network was much looser, and the internal space can accommodate more external moisture enters, leading to the swelling ratio being increased with the decreasing gelatin concentration. Even though the number of gelatin molecules reduced with decreased concentration, the gel network formed through TGase was still a compact structure that was maintained by covalent bonds, which still kept the ability to prevent the internal water from escaping when the gel was in a centrifugal state. Therefore, there was no significant difference (*p* < 0.05) in the WHC of TGase-induced gels.

During the heat-induced gelation, the filled emulsion droplets were flocculated and the spherical WPI molecules were clustered close to form a granular gel, while the linear gelatin molecules were spontaneously arranged triple helix and changed into a gel state [28]. Under the interaction of emulsion and matrix, a gel network formed and had smaller pores than the TGase-induced gels (shown in Figure 5) and was also accompanied by large uneven oil droplets occupying the space that should be the protein network, which made it difficult for external water to enter the internal, causing a lower swelling ratio. Even if the gelatin was increased, there was not much influence on the flocculation of oil droplets and aggregation of WPI because the gel behavior of WPI was more like the pure WPI gel, thus the structure of the final gel was basically similar [27]. It was still difficult for external water to enter, but increasing gelatin made the gel network stronger and firmer, which caused the resistance of the internal water to centrifugation force to increase and make the gel retain the original moisture inside the system better.

## 3. Conclusions

This study designed and constructed an emulsion-filled gel based on WPI-gelatin, and systematically explored the factors that may affect the texture properties of the gels. Heat and TGase treatment exhibited distinct formation mechanisms and macroscopic properties of the gels. For the heat-induced gels, the emulsions aggregated a lot, and the matrix network was discontinuous, while the pores were small but dense. For the TGase-induced gels, the emulsions dispersed uniformly without aggregation and with the continuous matrix networks and thick pore walls. The difference in microstructure made the TGase-induced gels have a larger elastic modulus and better swelling properties, and the heat-induced gels had a softer, more viscous texture and better water holding capacity. The change of emulsion droplet size and matrix concentration could endow the gels with various elastic modulus and hardness without affecting the interactions among the gels internal structure. This study provided a universal design idea for the tunable texture of composite emulsion gels, which contributed to the development and innovation of new gel-like healthy foods.

## 4. Materials and Methods

### 4.1. Materials

Whey protein isolate (WPI) was purchased from Shanghai Prochin International Trading Co., Ltd. (Shanghai, China). Gelatin (Type A, powder, gel strength ~300 g Bloom) was obtained from Sigma-Aldrich (Shanghai, China) Trading Co., Ltd. (Shanghai, China). Transglutaminase (TGase) was donated by Jiangsu Yiming Biological Co., Ltd. (Taixing, China). Corn oil was purchased from a local supermarket (Xianyang, China). All other reagents used were of analytical grade.

### 4.2. Methods

#### 4.2.1. Emulsion Preparation

WPI was selected as the emulsifier to prepare the oil-in-water emulsion, in which the volume fraction of oil phase was 20% *v*/*v*, and different concentrations of WPI (0.5–2 wt%) were prepared to select the appropriate concentration of emulsifier. Firstly, corn oil was poured into the WPI solution; after high-speed shearing (10,000 rpm, 5 min), the emulsion was prepared by high pressure homogenization. In order to obtain emulsions with different droplet sizes, homogenization was carried out by adjusting the homogenization pressure (5, 10, 50 MPa) [25].

#### 4.2.2. Preparation of Pre-Gel Mixture

The pre-gel mixture was composed of the WPI-gelatin mixture and emulsions, and the specific formulation is shown in Table 3. The different concentrations of protein dispersion were formulated, in which WPI was 6 wt%, and gelatin was 1–4 wt%, then blended with a volume ratio (VWPI:Vgelatin = 2:1) and stirred at 50 °C for 15 min to obtain the uniform mixture. Next, a certain volume of emulsion was poured into the protein mixture to obtain the pre-gel mixture, stirred evenly and set aside [34].

#### 4.2.3. Preparation of Emulsion-Filled Gels

Heat treatment (physical method): The pre-gel mixture was prepared according to procedure described in Section 4.2.2, then it was heated at 85 °C for 30 min in water bath, cooled to room temperature by running water immediately and stored overnight at 4 °C.

TGase induction (enzymatic method): WPI was preheated (85 °C, 20 min) in a water bath with gentle agitation, cooled to room temperature at once and then the pre-gel mixture was prepared according to procedure described in Section 4.2.2 [41]. TGase solution (80 U/g pro) was also added to the mixture and stirred evenly, then the mixture was incubated at 50 °C for 1 h and stored overnight at 4 °C.

Different emulsion gels samples were prepared according to the steps described above. The gel without the emulsion was taken as the emulsion blank gel, which was the gel formed only by gel continuous phase. Four factors which may affect gels performance (rheological and textural properties) were investigated, including treatment methods, emulsion filling volume (0–15% *v*/*v*), emulsion droplet size (depending on the homogenization pressure) and gelatin concentration (1–4 wt%).

#### 4.2.4. Characterization of Emulsions

The droplet size, polydispersity index (PDI) and zeta potential of emulsions were determined by dynamic light scattering and electrophoretic mobility measurements (Zetasizer Nano ZEN3600; Malvern Instruments Ltd., Malvern, UK). To reduce the multiple scattering effect, each sample was diluted with deionized water before measurement [24].

#### 4.2.5. Rheological Properties of Emulsion Gels

The rheological properties of emulsion gels were measured using a rotary rheometer (DHR-1 rheometer, TA Instruments, Waters, UK). A parallel aluminum foil plate with a diameter of 40 mm was selected in the clamp and the gap was set to 1 mm. A thin piece of gels was placed on the carrier platform, then a small amount of silicone oil was applied to its peripheral edge to prevent water evaporation. The 1% strain was identified to be in the linear viscoelastic region through the strain scanning (data not shown). Storage modulus (G′) and loss modulus (G″) were observed under frequency scanning mode with frequency ranges from 0.1 to 100 rad/s at 1% strain. The flowing scanning test was carried out to obtain the relationship between apparent viscosity and shear rate ranging from 0.1 to 100 s^−1^ [32]. Each sample was measured in triplicate.

#### 4.2.6. Texture Properties of Emulsion Gels

Texture profile analysis (TPA), a test consisted of two continuous compression cycles, was carried out by a Texture Analyzer (TA.XT PLUS/50, STABLEMICVO, Godalming, UK) using a P/0.5R cylindrical probe. The compression distance was 10 mm, triggering force was 5 g, predicted test speed was 1.5 mm/s, test speed was 1.0 mm/s, and the post-test speed was 1.0 mm/s too [32]. Each sample was repeated three times.

Stress relaxation test: After overnight storage at 4 °C, emulsion gels were completely removed from the container. At room temperature, gel was compressed to 10% of its initial height at a constant speed of 1.0 mm/s using a P100 probe, and the relaxation time lasted for 60 s after reaching a given strain, meanwhile the change of the stress on the gel under a fixed strain was recorded.

#### 4.2.7. Sodium Dodecyl Sulfate Polyacrylamide Gel Electrophoresis (SDS-PAGE)

SDS-PAGE was used to investigate the mechanism of emulsion gels formation under different crosslinking conditions. The protein concentration of samples was diluted to 2 mg/mL with deionized water and mixed evenly with the buffer solutions on the reduction (including β-mercaptoethanol) and non-reduction (without β-mercaptoethanol), so the final protein concentration was 1 mg/mL [42]. It should be noted that reduction group was boiled for 5 min and non-reduction group was left untreated. The concentration of separating gel was 12.5% and stacking gel was 4%. The loading volume of each lane was 10 μL, and the constant voltage was 120 V. After electrophoresis, the gel was stained with Coomassie brilliant blue R250, then decolorized with a mixture of glacial acetic acid and methanol, and finally photographed and recorded [7].

#### 4.2.8. Confocal Laser Scanning Microscope (CLSM)

The oil phase and continuous phase distribution of emulsion gels were observed by CLSM. Oil droplets and proteins were stained with Nile red and Fast Green FCF, respectively. Emulsion gels were cut to a thin piece and placed on a concave slide. Next, 10 μL Nile red and Fast Green FCF were added for staining, then covered the slide and the images were taken at excitation wavelength of 488 nm and 633 nm [24].

#### 4.2.9. Cryo-Scanning Electron Microscope

The microstructure of emulsion gels was observed through a cold field emission scanning electron microscope equipped with a low temperature transfer printing system. A small piece of the emulsion gels was placed on a copper sample pile coated with a conductive glue. After undergoing low-temperature cooling fixation, freeze-fracture and sublimation in sequence, the sample pile was sent to the cold stage in the machine through the frozen transfer sample rod, and then the observation was performed.

#### 4.2.10. Hydration Properties of Emulsion Gels

Water holding capacity. Water holding capacity (WHC) of emulsion gels was determined by centrifugation treatment. Firstly, 2 g of wet gels were weighed and placed in a centrifugal tube, then centrifuged for 15 min at room temperature at 8000 rpm; subsequently, the precipitated water was removed by filter paper and the gel was weighed again. WHC was calculated according to Formula (2) below:(2)$$\mathrm{WHC}=\frac{w_{2}-w}{w_{1}-w}\times 100\%$$
where W_2_ was the total weight of the gels and tube after centrifugation (g); W_1_ was the initial total weight of the gels and tube (g); W was the tube weight (g). Each sample was measured in triplicate.

Swelling ratio. The swelling behavior of emulsion gels was investigated according to the previous research with slight modification [43]. An amount of 3 g of emulsion gel was soaked in distilled water for 24 h to achieve the swelling equilibrium and reweighted. The swelling ratio was numerically equal to the percentage ratio of the mass difference before and after swelling to the initial weight.

#### 4.2.11. Data Processing and Analysis

The raw data obtained in the experiment was used to calculate the mean and standard deviation with the help of Excel of Microsoft Office 2021. Minitab v.18.1 statistical software was used to calculate the Pearson correlation coefficient and analysis of variance. All data graphs were drawn using Origin pro2022SR1. In analysis of variance, Fisher’s LSD test was used for multiple comparisons and data was considered to be significantly different (*p* < 0.05).

## Figures and Tables

**Figure 1 gels-08-00212-f001:**
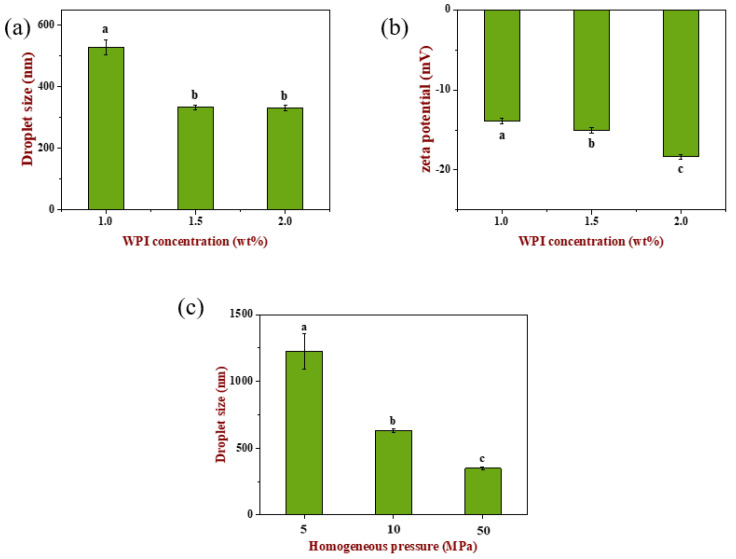
Optimization of the oil-in-water emulsion stabilized by WPI as emulsifier. (**a**) The effect of WPI concentration on the droplet size of the emulsion; (**b**) the effect of WPI concentration on the zeta potential of the emulsion; (**c**) the effect of homogenization pressure on the droplet size of the emulsion. Samples designated with different lower-case letters indicate significant differences (*p* < 0.05) when compared between different samples.

**Figure 2 gels-08-00212-f002:**
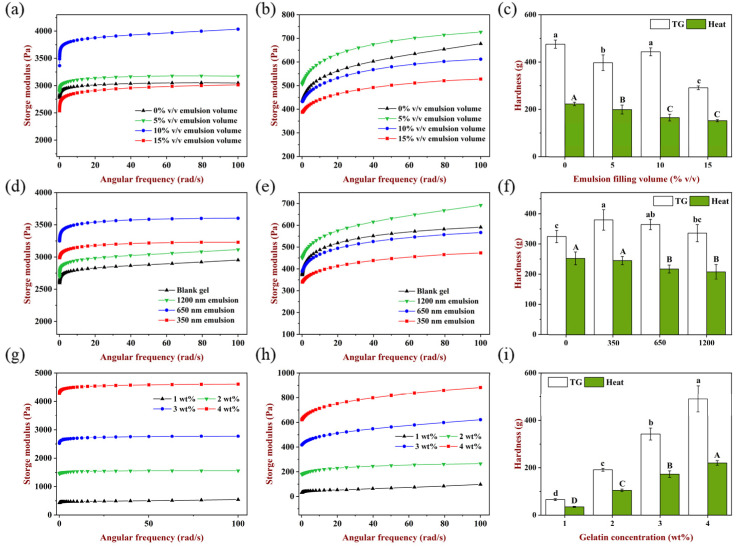
Effect of emulsion filling volume on the storage modulus of (**a**) TGase-induced gels, (**b**) heat-induced gels and (**c**) hardness of emulsion gels. Effect of emulsion droplet size on the storage modulus of (**d**) TGase-induced gels, (**e**) heat-induced gels and (**f**) hardness of emulsion gels. Effect of gelatin concentration on the storage modulus of (**g**) TGase-induced gels, (**h**) heat-induced gels and (**i**) hardness of emulsion gels. ‘TG ‘in the legend of the column graph refers to TGase. Samples designated with different letters indicate significant differences (*p* < 0.05) when compared between different samples.

**Figure 3 gels-08-00212-f003:**
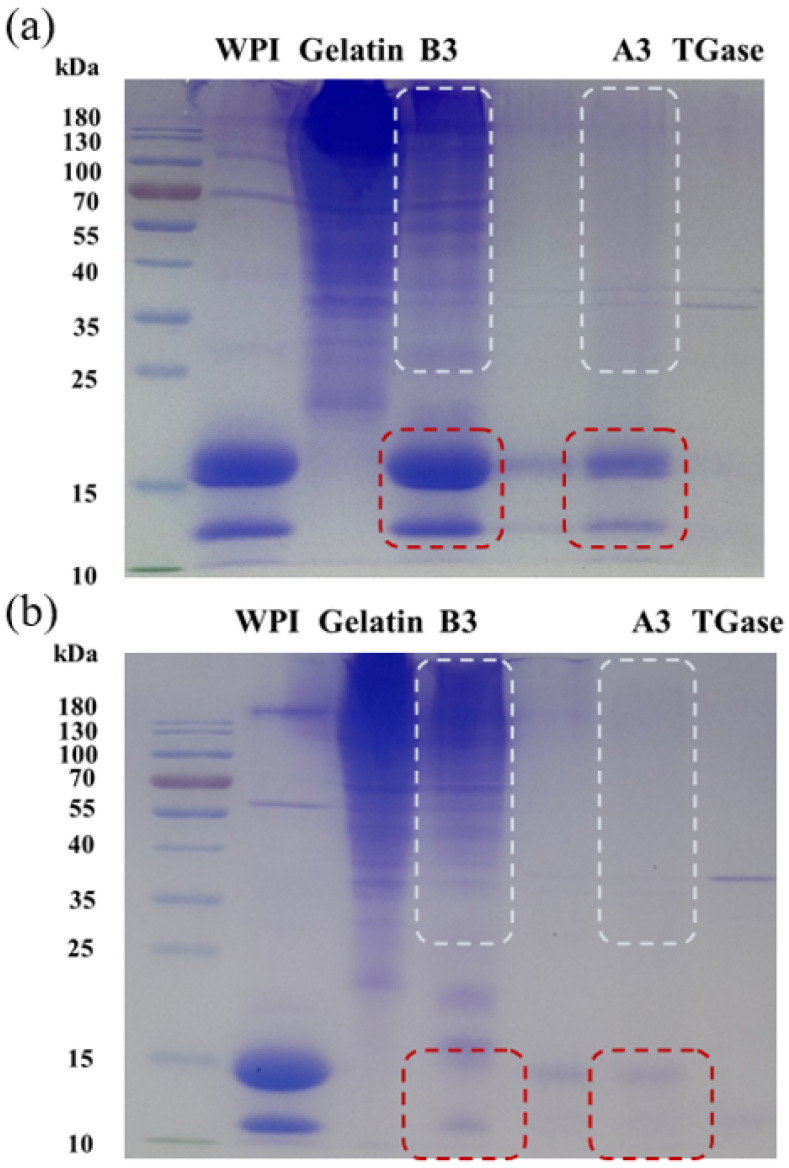
The SDS-PAGE of (**a**) reduction and (**b**) non-reduction of emulsion gels.

**Figure 4 gels-08-00212-f004:**
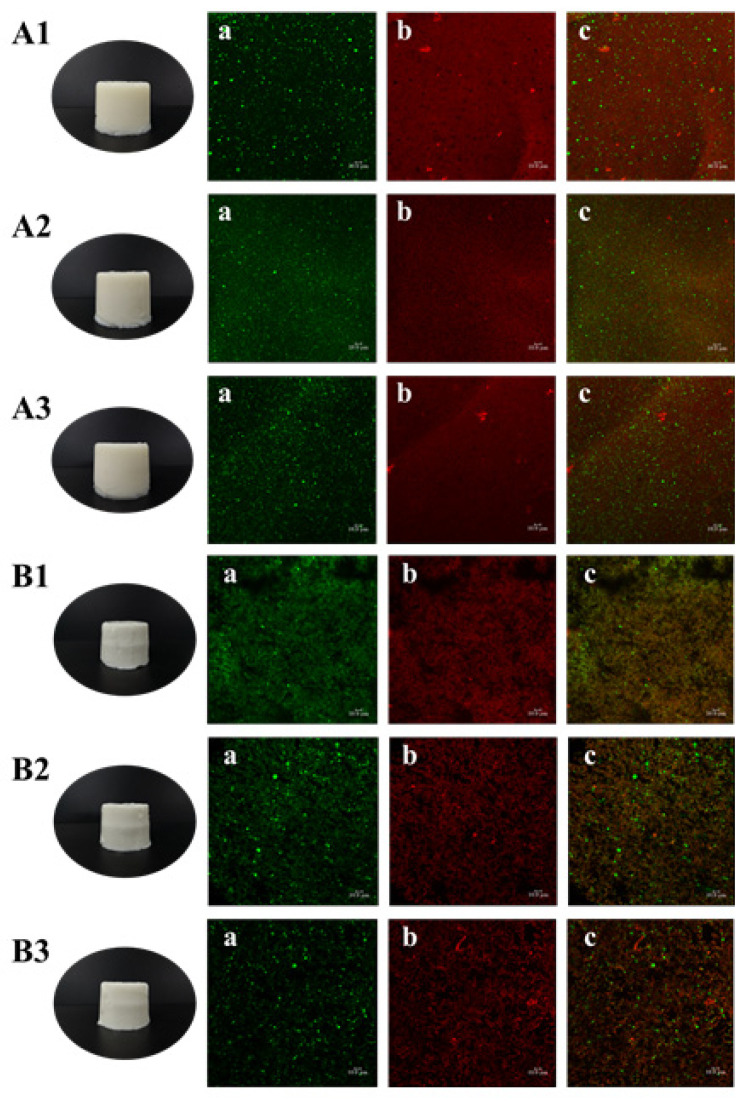
Appearance and confocal laser scanning microscope image (40×) of (**A**) TGase-induced gels, where (**A1**) means the gel with 2 wt% gelatin and 1200 nm emulsion, (**A2**) means the gel with 3% gelatin and 350 nm emulsion and (**A3**) means the gel with 4 wt% gelatin and 650 nm emulsion, (**B**) heat-induced emulsion gels, where (**B1**) means the gel with 2 wt% gelatin and 350 nm emulsion, (**B2**) means the gel with 3% gelatin and 650 nm emulsion and (**B3**) means the gel with 4 wt% gelatin and 1200 nm emulsion.

**Figure 5 gels-08-00212-f005:**
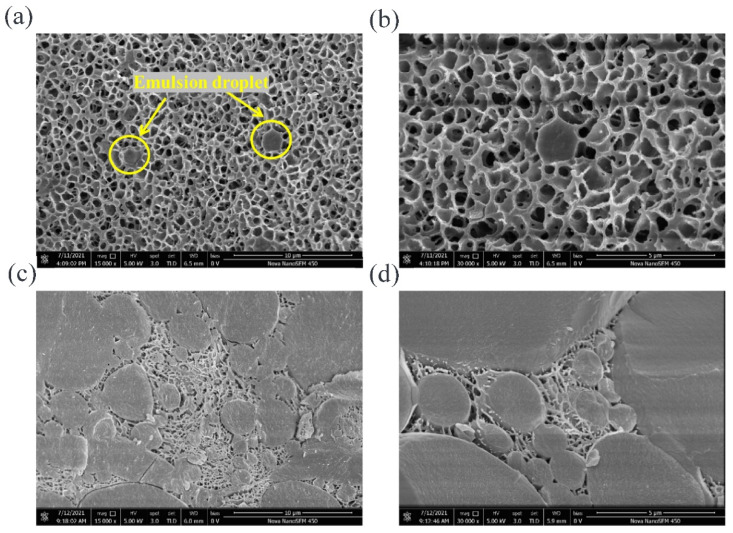
Cryo-SEM images of (**a**) 15,000× and (**b**) 30,000× magnification of TGase-induced emulsion gels, (**c**) 15,000× and (**d**) 30,000× magnification of heat-induced emulsion gels.

**Figure 6 gels-08-00212-f006:**
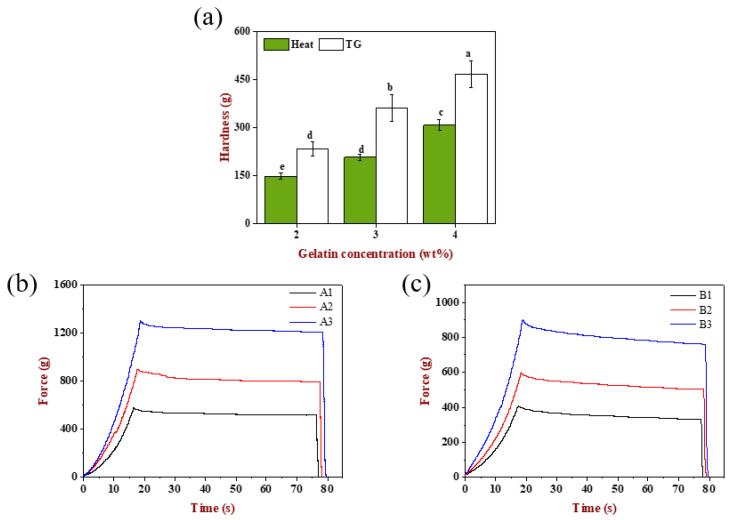
(**a**) Hardness of TGase-induced and heat-induced emulsion-filled gels. The stress-relaxation curves of (**b**) TGase-induced and (**c**) heat-induced emulsion-filled gels. Samples designated with different letters indicate significant differences (*p* < 0.05) when compared between different samples.

**Figure 7 gels-08-00212-f007:**
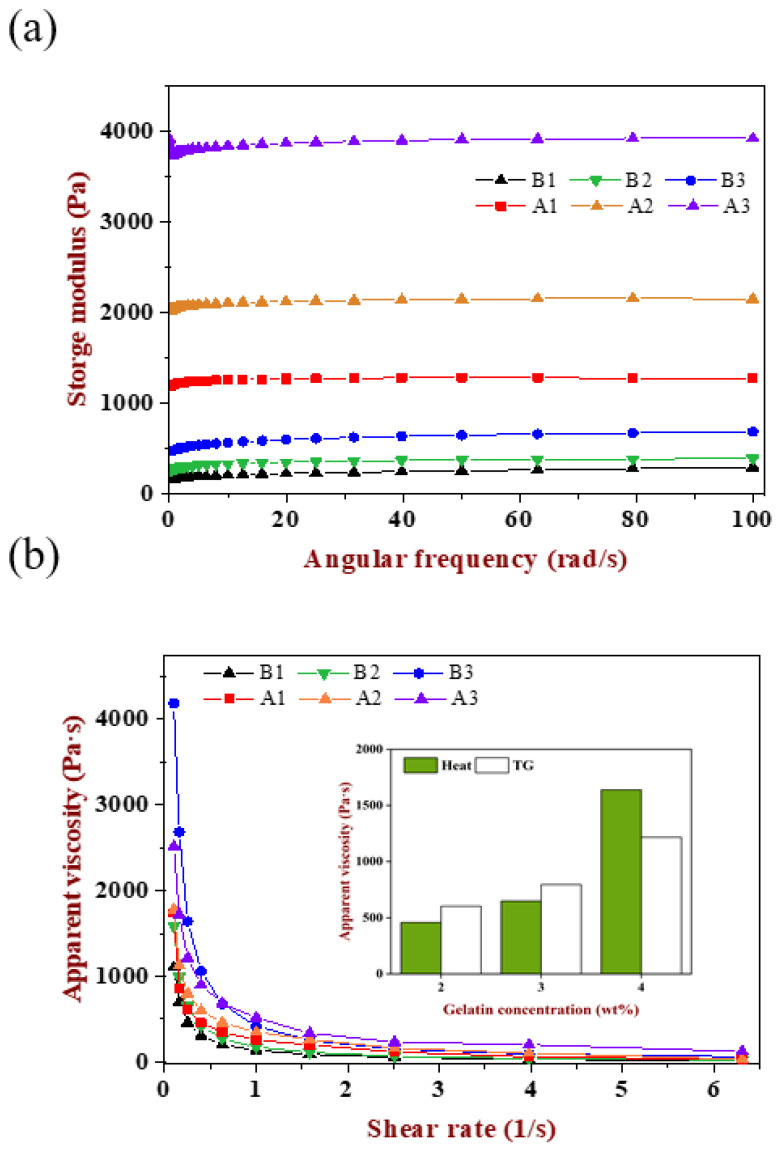
(**a**) Frequency sweep and (**b**) flow sweep of emulsion-filled gels, the internal column graph was the apparent viscosity of emulsion gels at a fixed shear rate at 0.25/s and the abscissa was differentiated according to the gelatin concentration.

**Figure 8 gels-08-00212-f008:**
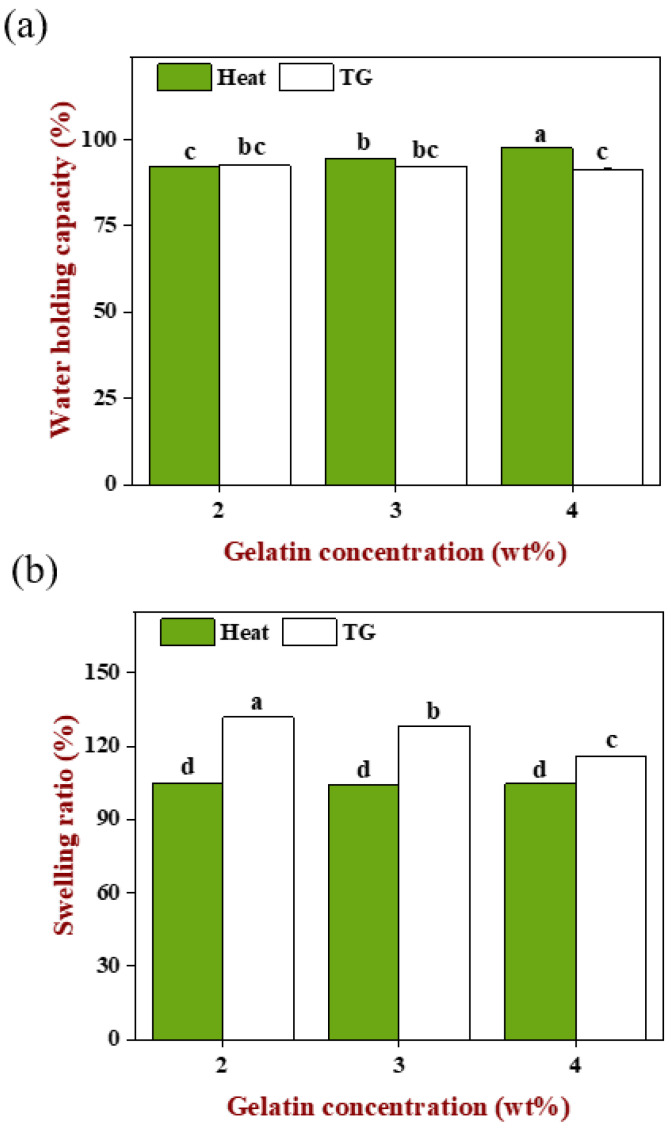
(**a**) WHC and (**b**) swelling ratio of emulsion-filled gels. Samples designated with different letters indicate significant differences (*p* < 0.05) when compared between different samples.

**Table 1 gels-08-00212-t001:** Pearson correlation (r) of three factors and hardness or storage modulus of emulsion-filled gels which formed by TGase induction and heat induction.

	TGase-Induced Emulsion Gels				Heat-Induced Emulsion Gels			
	Hardness		Storage Modulus		Hardness		Storage Modulus	
	r	*p*	r	*p*	r	*p*	r	*p*
Emulsion-filled volume	−0.814	0.186	0.171	0.829	−0.987	0.013	−0.778	0.222
Emulsion droplet size	−0.999	0.029	−0.421	0.723	−0.919	0.126	1.000	0.014
Gelatin concentration	0.999	0.001	0.991	0.009	0.996	0.004	0.995	0.005

Pearson correlation (r) > 0 represents the relationship is a positive correlation.

**Table 2 gels-08-00212-t002:** Three-element Maxwell model fitting parameters.

	E_0_ (N/mm)	E_1_ (N/mm)	T (s)	η (N·s/mm)	R^2^
A1	62.95078	7.322125	11.90356	87.15935427	0.98
A2	90.64707	11.93894	12.00635	143.3430923	0.98
A3	129.9613	9.664588	11.95991	115.5876027	0.96
B1	38.41167	9.145636	15.4657	141.4436627	0.97
B2	55.27848	10.6386	16.17971	172.1294628	0.96
B3	81.06408	15.12954	16.57686	250.8002664	0.96

**Table 3 gels-08-00212-t003:** The composition of pre-gel mixture.

	WPI	Gelatin	Emulsion
Pre-gel mixture	6 wt%	1–4 wt%	0–15% *v*/*v*

## Data Availability

Not applicable.

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
