# Peer review of "Comparative Study of Heat- and Enzyme-Induced Emulsion Gels Formed by Gelatin and Whey Protein Isolate: Physical Properties and Formation Mechanism"

_gels, 2022, doi:10.3390/gels8040212_

Round 1

Reviewer 1 Report

The aim of this work is to prepare emulsion gels based on whey protein isolate and gelatin by two different methods: an enzymatic and a physical method. The obtained emulsion gels are well characterized in terms of their structure and rheological and mechanical properties. Droplet size and gelatin concentration are the factors that most affect the structure and texture of emulsion gels..The results of this study are important for structuring emulsion gels and adjusting the texture of food products. The manuscript is presented in a well-structured manner.
The references cited are relevant and current, mostly published within the last three years.
The results of the manuscript are reproducible using the information in the methods section, but some minor explanations are missing. What equipment was used for the texture profile analysis? Describe the emulsion blank gel. What does the term "wet gel" mean? 
All figures and tables are well presented. 
The conclusions are consistent with the presented results of the study.

Author Response

Point 1: The aim of this work is to prepare emulsion gels based on whey protein isolate and gelatin by two different methods: an enzymatic and a physical method. The obtained emulsion gels are well characterized in terms of their structure and rheological and mechanical properties. Droplet size and gelatin concentration are the factors that most affect the structure and texture of emulsion gels. The results of this study are important for structuring emulsion gels and adjusting the texture of food products. The manuscript is presented in a well-structured manner. The references cited are relevant and current, mostly published within the last three years. The results of the manuscript are reproducible using the information in the methods section, but some minor explanations are missing.

 Response 1: It is our great pleasure to get your positive comments. Thanks very much for your responsible and efficient work in the process of our manuscript.

 Point 2: What equipment was used for the texture profile analysis?

 Response 2: Thanks for your suggestion. In this study, texture profile analysis was investigated by a Texture Analyzer (TA.XT PLUS/50, STABLEMICVO, UK) using a P/0.5R cylindrical probe, and corresponding description has been added to the manuscript in Line 526 marked in red font.

Point 3: Describe the emulsion blank gel.

Response 3: Thanks for your comment. The gel without the emulsion was taken as the emulsion blank gel, which was the gel formed only by gel continuous phase.

Point 4: What does the term "wet gel" mean?

Response 4: Thanks for your question. Wet gel were the emulsion gels we obtained, which had a three-dimensional network structure that can hold a large amount of water. The corresponding Line  570 in the manuscript has been revised to a more accurate expression, emulsion gels. Thanks again for your careful review.

Reviewer 2 Report

This article compares the production of emulsion gels by two different methods, namely heat-and TGase- induced emulsion gels. The article is very well written and only requires few minor changes. One main request would be to make it clearer what was the formulations and conditions tested.

Line 60: define WPI at first use

Line 84-88: This is a summary of what was carried out and not a discussion, please remove

Figure 1: Include in Fig.1c what was the the % of WPI used

Figure 2: Increase the font of the legend

Line 166: English check

Figure 3: rename using latin alphabet

Figure 4: Increase legend as it is not visible. B1, B2, and B3 looks biphasic, was this the case?

Figure 7-B: I suggest choosing just one shear rate, instead of providing both graphs one can provide the numerical values at a fixed shear rate

Line 474: What is meant by optimize condition?

Line 478 and 485: Are the gel continuous phase and the protein-emulsion mixture the same thing? This section is a bit confusing so I would suggest authors include a Table with all the sample formulations/conditions tested.

Line 496: remove respectively

Line 499. Characteristics or characterization?

Line 500: Define PDI at first use

Line 507: Clearance height or gap`?

Line 512: Was 1% found to be within the viscoelastic region?

Line 516: rephase

Line 530: English check

Author Response

Point 1: This article compares the production of emulsion gels by two different methods, namely heat-and TGase- induced emulsion gels. The article is very well written and only requires few minor changes. One main request would be to make it clearer what was the formulations and conditions tested.

 Response 1: It is our great pleasure to get your professional comments. Thanks very much for your contribution to our manuscript. We take the concerns seriously and comments have been carefully addressed.

Point 2: Line 60: define WPI at first use.

Response 2: Thanks for carefully checking. Whey protein isolate (WPI) has been added in Line 60-61 in the manuscript.

Point 3: Line 84-88: This is a summary of what was carried out and not a discussion, please remove.

Response 3: Thanks for your constructive comment. Line 84-88 has been removed and it made the manuscript more logically. Thank you again for the good comment.

Point 4: Figure 1: Include in Fig.1c what was the the % of WPI used.

Response 4: Thanks for your comment. The emulsions of different droplet size was prepared at WPI concentration of 1.5 wt% and this information has been added in Line 102. This concentration was selected as a suitable emulsifier concentration based on the results in Figure 1a and 1b, and more details can be found in the manuscript.

Point 5: Figure 2: Increase the font of the legend.

Response 5: Thanks for your good comment. Figure 2 has been modified accordingly to make the legend clearer.

Point 6: Line 166: English check.

Response 6: Thanks for your carefully check. The manuscript has been revised accordingly in Line 159-161.

Point 7: Figure 3: rename using latin alphabet

Response 7: Thanks for your carefully check. Figure 3 has been revised accordingly.

Point 8: Figure 4: Increase legend as it is not visible. B1, B2, and B3 looks biphasic, was this the case?

Response 8: Thanks for your comment. Figure 4 has been adjusted to make the legend clear. In fact, B1, B2 and B3 were in a relatively homogeneous state, and no phase separation had occurred. The shadow in the middle part of the apparent image of the heat-induced gels was because the texture test was carried out before shooting. Unlike TGase-induced emulsion gel had very good elastic texture, the structure of heat-induced gels did not fully recover to its original state due to its soft and less elastic texture after compression.

Point 9: Figure 7-B: I suggest choosing just one shear rate, instead of providing both graphs one can provide the numerical values at a fixed shear rate.

Response 9: Thanks for your constructive suggestion. Figure 7B has been revised accordingly, that is, a graph was added which displayed the apparent viscosity of all emulsion gels at a fixed shear rate of 0.25/s, showing the differences between groups very visually. Thanks again for your suggestion.

Point 10: Line 474: What is meant by optimize condition?

Response 10: Thanks for your question. In order to choose appropriate emulsifier, we prepare different WPI concentration of 1.0-2.0 wt% and found the emulsion formulated with 1.5 wt% WPI had smaller size and medimum zeta potential, so it was described to the optimize condition. For clarity, we have also revised the corresponding positions in the manuscript.

Point 11: Line 478 and 485: Are the gel continuous phase and the protein-emulsion mixture the same thing? This section is a bit confusing so I would suggest authors include a Table with all the sample formulations/conditions tested.

Response 11: Thanks for your good suggestion. The gel continuous phase was the WPI-gelatin mixture. To more clearly describe the composition of the emulsion gel, Table 3 has been added to the manuscript. The pre-gel mixture, refered to the mixture before gelation occurred, was composed of WPI-gelatin mixture and emulsions, but due to the large number of samples prepared by the single factor experiment, only the general formulation of the emulsion gel was listed in Table 3. And the relevant statements in 4.2.2 and 4.2.3 have been revised in the manuscript.

Point 12: Line 496: remove respectively

Response 12: Thanks for your check. It was revised in the manuscript.

Point 13: Line 499. Characteristics or characterization?

Response 13: Thanks for your carefully review. The characterization of emulsions was investigated by the droplet size, polydispersity index (PDI) and zeta potential. This expression has been revised in the manuscript.

Point 14: Line 500: Define PDI at first use

Response 14: Thanks for your comment. PDI was the polydispersity index and the manuscript has been revised in Line 501.

Point 15: Line 507: Clearance height or gap`?

Response 15: Thanks for your comment. The gap was set to 1 mm. Thank you again for your careful review.

Point 16: Line 512: Was 1% found to be within the viscoelastic region?

Response 16: Thanks for your question. 1% strain was within the linear viscoelastic region in our pre-experiment so it was chosen for other tests.

Point 17: Line 516: rephase

Response 17: Thanks for your comment. The statement in 4.2.5 has been revised in the manuscript.

Point 18: Line 530: English check

Response 18: Thanks for your carefully check. Line 533 has been modified which corresponding to the line of revised manuscript.